# Effects of Visible Light on Gas Sensors: From Inorganic Resistors to Molecular Material-Based Heterojunctions

**DOI:** 10.3390/s24051571

**Published:** 2024-02-29

**Authors:** Sujithkumar Ganesh Moorthy, Marcel Bouvet

**Affiliations:** Institut de Chimie Moléculaire de l’Université de Bourgogne (ICMUB), UMR CNRS 6302, Université de Bourgogne, 9 Avenue Alain Savary, 21078 Dijon CEDEX, France; sujithkumar_ganesh-moorthy@etu.u-bourgogne.fr

**Keywords:** conductometric transducers, resistors, heterojunctions, light effect, adsorption–desorption mechanism

## Abstract

In the last two decades, many research works have been focused on enhancing the properties of gas sensors by utilising external triggers like temperature and light. Most interestingly, the light-activated gas sensors show promising results, particularly using visible light as an external trigger that lowers the power consumption as well as improves the stability, sensitivity and safety of the sensors. It effectively eliminates the possible damage to sensing material caused by high operating temperature or high energy light. This review summarises the effect of visible light illumination on both chemoresistors and heterostructure gas sensors based on inorganic and organic materials and provides a clear understanding of the involved phenomena. Finally, the fascinating concept of ambipolar gas sensors is presented, which utilised visible light as an external trigger for inversion in the nature of majority charge carriers in devices. This review should offer insight into the current technologies and offer a new perspective towards future development utilising visible light in light-assisted gas sensors.

## 1. Introduction

Gas sensors play a pivotal role in monitoring and detecting the presence of specific gases in various applications, ranging from environmental monitoring to industrial safety and healthcare. The need for more accurate, sensitive and selective gas sensors has led to ongoing research in this field. The ability to detect and quantify specific gases or volatile organic compounds is essential for ensuring safety, quality and compliance [1,2]. Traditionally, gas sensors have relied on well-established principles of chemical interactions, which are inherently affected by factors like temperature, humidity and the presence of interfering gases [3,4].

In recent years, an emerging and fascinating approach that has garnered considerable attention is the combination of light sources with gas sensors [5,6]. Light-assisted techniques have offered a transformative approach to address the limitations of traditional gas-sensing technologies and harness the power of illumination to influence and enhance their gas-sensing capabilities [7,8]. Researchers worldwide have been instrumental in advancing this technology by utilising various forms of electromagnetic radiation, such as ultra-violet (UV) [9,10,11,12], visible light [13,14], and infra-red (IR) radiations [15] to improve gas-sensing performances. These sensors have gained significance due to their ability to provide high sensitivity, short response times and low power consumption in gas detection, at least with visible light, while also expanding their applicability in various environments. However, the effect of visible light on the gas sensors has gained more attention than any other light because of its low cost and low power consumption [16]. Moreover, utilising visible light as an activation source offers the advantage of making various light sources commonly available in everyday life suitable for light-activated gas-sensing technologies. This practical approach broadens the range of applicable light sources to those accessible throughout a person’s lifetime. Moreover, employing visible light sources serves to avoid potential harm to sensing materials that may occur with UV light, with the dual purpose of reducing the thermal effect and ozone generation in oxygen-rich environments [17].

In this review article, we reviewed the visible light effect on chemoresistors and heterojunction gas sensors and tried to give a clear understanding of the involved phenomena. In particular, the fascinating concept of ambipolar gas sensors is presented, which utilized visible light as an external trigger for inversion in the nature of majority charge carriers in devices [18].

## 2. Fundamentals and Mechanisms of Light-Activated Gas Sensors

Gas sensors are founded upon the fundamental principle that they operate by recognising the changes that occur when target gases interact with a sensing material. This specific interaction leads the gas molecules to adsorb or absorb onto the sensor’s surface or within its material structure, respectively. Depending on the type of sensor and the gas involved, sorption behaviour can lead to changes in the sensor’s electrical, optical or thermal properties based on the concentration of the target gas [19]. Understanding and controlling sorption behaviour is crucial for designing effective gas sensors with high sensitivity and selectivity [20,21].

Light irradiation upon the gas sensors effectively modifies the sorption behaviour and improves its kinetics due to the photoexcitation of charge carriers in semiconductors [9]. It is well known that when light energy is absorbed by gas molecules or sensing materials, it leads to electronic transitions [22,23,24]. In simpler terms, photons (hν), or packets of light energy, knock electrons from their stable positions, creating both negatively charged electrons $\left(e^{-}\right)$ and positively charged holes $\left(h^{+}\right)$ in conduction and valence energy bands, respectively, when inorganic semiconductors are considered (Figure 1).

Semiconductors have a specific energy gap between their valence and conduction bands or highest occupied molecular orbital and lowest unoccupied molecular orbital in molecular semiconductors, known as the bandgap or molecular bandgap [25,26]. The excitation of electrons to higher energy levels alters the conductivity of the material via the change in free charge carrier density. But more interesting is the increase in the relative response (RR), which can even depend on the wavelength. Thus, with carbon nanotubes, it was shown that the RR was multiplied by a factor of 1.5 when changing from 365 nm (0.42–0.55 mW cm^−2^) to 275 nm (8–10 mW cm^−2^), while the response was barely visible in the dark [27].

In further depth, the interaction between light and gas sensors is rooted in the principles of photochemistry and semiconductor physics [28]. It involves a series of reactions initiated by the absorption of photons [29,30]. Most conductometric devices are sensitive to light [31,32]. In the case of dye-sensitized semiconductors, e.g., perylenediimide/SnO_2_ heterojunction device, light absorption generates e^−^ that are injected in the conduction band of the semiconductor, increasing its conductivity; meanwhile, the generated holes can recombine with adsorbed $O_{2}^{-}$ ions, leading to *O*_2_ formation, facilitating its desorption (Equation (1)). In this case, the choice of the dye allows the use of a lower energy light compared to what is needed to excite the semiconductor [31]. Another key effect of light is its impact on gas desorption [33], often of *O*_2_ in atmospheric conditions, but not only oxygen.
(1)$$h^{+}+O_{2(ads)}^{-}\to O_{2(ads)}$$

In addition to photo-stimulated desorption, photo-stimulated adsorption was also mentioned in the case of MoS_2_-based NO_2_ sensors [34]. This could be due to the increase in free adsorption sites related to oxygen desorption under illumination. Meanwhile, the oxygen in the surroundings reacts with the photo-induced electrons, generating additional photo-induced oxygen ions (Equation (2)) [10].
(2)$$O_{2(ads)}+e^{-}\overset{hv}{\to }O_{2(ads)}^{-}$$

So, in general, light absorption modifies the adsorption–desorption equilibrium and can explain why recovery time can be reduced [35].

Initially, the idea of light activation on gas sensing was ignited by D. A. Melnick. He explored the mechanism of oxygen adsorption and its impact on the conductivity of porous sintered zinc oxide samples, particularly when exposed to UV light [36]. This study involved examining the photoconductance of zinc oxide samples, wherein their electrical conductivity changed upon exposure to light. It is observed that the conductivity increases during illumination and decreases when the light source is turned off. In ZnO, which is an n-type semiconductor, chemisorbed oxygen acts as a trap. So, under illumination, oxygen desorption leads to a conductivity increase. Photo-stimulated adsorption and desorption of oxygen were also evidenced in the case of SnO_2_ and TiO_2_ [24].

In 1962, T. Seiyama introduced the first ever metal oxide-based gas sensors working at high temperatures, over 400 °C, detecting several gases, including toluene, benzene, CO_2_, propane and ethyl ether [37]. This effect drew the attention of many researchers to use an external trigger to improve or enhance the sensing properties of gas sensors. Performance-wise, these gas sensors, which operate at high temperatures, certainly secured fast kinetics with superior sensing responses towards various gases. However, the increasing in demand for portable and multifunctional devices led the researchers to focus on reducing power consumption by lowering the operating temperature of the gas sensors. J.T. Cheung demonstrated and patented the light-assisted gas sensors in the 1990s [38], demonstrating the interest of light irradiation for stimulation of desorption of gas from the surface of the sensing material at room temperature. The approach of using light in gas sensors ignited the light-assisted gas sensing technology, resulting in numerous scientific publications in the following years.

## 3. Enhancement in Gas-Sensing Properties of the Sensors under Visible Light Illumination

Light radiations like IR and UV can significantly increase the temperature of gas sensors, which can lead to severe damage to the active sensing material, depending on wavelength and illumination power. Another effect that occurs with high energy light is the generation of ozone, most of gas applications being in real atmosphere [17]. This point is very important when sensing materials are carbonaceous materials or molecular materials. Indeed, ozone is a very strong oxidising agent that reacts easily with C–C double bonds [39]. However, ozone formation is supposed to occur at a wavelength of 240 nm. In real environmental sensor applications, these issues can be overcome by replacing UV light by utilising visible light as an external trigger or activation source.

Various approaches like doping, compositing or sensitising impurities with single materials, hybrids or heterostructures and light irradiation have been studied to reduce operating temperatures and enhance the sensitivity and stability of gas sensors. Many research papers reported a light effect on inorganic material-based chemoresistors and heterojunction devices as developed in recent reviews [40,41,42,43,44]. Herein, we only describe typical examples.

### 3.1. Chemoresistors

Chemoresistors are nothing but conductometric sensors made up of only one layer unlike heterostructure sensors that consist of two or more different types of materials that form junctions [45]. Over the decades, many scientists have been working on various metal oxide semiconductor materials in the field of gas-sensing applications and trying to observe their novel behaviour, both in the dark and under light. Recent studies show that the visible light illumination on gas sensors based on metal oxides like zinc oxide (ZnO), tin dioxide (SnO_2_), tungsten oxide (WO_3_) and indium oxide (In_2_O_3_) has emerged as a prominently researched method for achieving high sensitivity at room temperature [46,47,48,49,50,51,52,53].

Q. Geng et al. report the enhancement of ZnO-based gas sensors under visible light illumination. In this work, they studied different reducing gases like ethylene and acetone in the dark and under white light, with different bandpass transmission filters (420, 450, 500 and 520 nm) [47]. ZnO-based sensors exhibit no sensing response towards both reducing gases (ethylene and acetone) in the dark. However, the sensors were instantly activated under visible light illumination, and distinct responses were apparent under each visible light irradiation. Figure 2a shows the response curves of a ZnO-based sensor in the dark and under visible light illumination, defined as the ratio Ra/Rg, where Ra and Rg are the responses in air and under gas, respectively. According to the authors, the activation of ZnO-based sensors under visible light can be depicted based on a two-photon or multi-photon excitation process. Visible light interaction induces a photo-induced desorption process of oxygen, increasing electron density at the ZnO surface, which is an n-type semiconductor. This process establishes a photo-induced adsorption–desorption cycle of oxygen (Figure 2b). Upon exposure to the target gas, the photo-induced desorption of O_2_ offers unoccupied adsorption sites, promoting the adsorption of other organic molecules. The chemisorbed organic molecule then contributes electrons to ZnO, enhancing the separation of photo-generated charges and increasing the ZnO surface electron density, thereby enhancing the sensitivity to ethylene and acetone.

Modifying the oxygen vacancy in sensing materials can highly impact the sensing properties of gas sensors [54]. When ZnO_1−x_ films with different degrees of oxygen vacancy were subjected to white light, they exhibited an increasing sensing response with an increasing concentration of oxygen vacancies. The coatings with the highest oxygen vacancy concentration, V_o_ = 4.8% (ZnO_0.952_), show a significant sensing response of 5.3% towards 0.9 ppm of NO_2_, with fast response/recovery time, 1.8/2.7 min, respectively. Based on some in situ spectroscopic studies, the enhanced NO_2_ sensing performance was apparently achieved from the co-modulation of surface chemisorption and reactions between the species of $O_{x}^{-}$ and $NO_{x}^{-}$ by light effect and oxygen vacancies. The photo-generated electrons were trapped by surface oxygen vacancies to enhance the ionised adsorption of O_2_ so as to produce highly active oxygen species $\left(O^{-}\right)$ and then trigger the $NO_{3}^{-}$ species chemisorption. A similar effect was observed with black NiO, a p-type semiconductor, under 480 nm blue light illumination [55]. Black NiO chemoresistor with the highest oxygen vacancy concentration experimentally exhibits RR of 11.6% towards 57 ppb of NO_2_. Therefore, the light irradiation on rich surface oxygen vacancies greatly increased the adsorption of NO_2_ and the electron transfer from ZnO to NO_2_, resulting in improved room temperature NO_2_ sensing properties.

Apart from ZnO, SnO_2_ is the most widely used material for metal oxide semiconductor-based gas sensors due to its high sensitivity and good stability ever since its integration into real-life applications, presented and patented by Taguchi in the 1960s [56]. Subsequent to initial research, Comini et al. studied the SnO_2_ sensor performance in detecting NO_2_ under UV illumination [57]. Their study shows a stable and highly responsive sensor operating at room temperature with UV excitation (λ = 365 nm) as an external trigger. It allowed them to achieve complete sensor recovery at room temperature with a response/recovery time of 15/4 min, respectively. More interestingly, Anothainart et al. achieved a similar sorption behaviour in SnO_2_-based NO_2_ gas sensors by replacing UV with polychromatic light [58]. They observed the accelerated desorption of NO_2_ from the SnO_2_ sensor by utilising a 12 W halogen lamp with a combination of different spectral and attenuation filters. Their work highlighted the effect of light with a wavelength below 600 nm and demonstrated the dependence of desorption on light intensity. By measuring the conductance and work function at both room temperature and elevated temperatures, they deduced the light-activated desorption occurred primarily due to the direct photoexcitation of electrons from $NO_{2}^{-}\;$ adsorbates into the conduction band of SnO_2_ rather than the generation of electron–hole pairs.

Different wavelengths and intensities of light can significantly impact the sensing response of the sensor. Under light illumination WO_3_-based NO_2_ sensor exhibits increasing in response from 2.7% to 9.2% towards 160 ppb concentration of NO_2_, with increasing in light wavelengths from 380 to 590 nm, respectively. Among different wavelengths, blue light (480 nm) with higher light intensity (370 mW cm^−2^) has a higher impact on the sensor’s response/recovery time, 14.6/18.3 min, respectively, compared to other light wavelengths and intensities [59]. Similar to this work, L. Deng et al. measured the conductivity of the mesoporous WO_3_ under formaldehyde upon both white and blue lights and compared it with commercially available WO_3_. Mesoporous WO_3_ has a higher absorbance of visible light because of the doped carbon in the crystal lattice [46,60]. Hence, the band gap of mesoporous WO_3_ is reduced to 2.12 eV compared to commercial WO_3_, which has a band gap of 2.45 eV. The response of the mesoporous WO_3_-based sensor to formaldehyde increases 4 times under white light (10 mW cm^−2^) and 9 times under blue light (10 mW cm^−2^), compared to commercial WO_3_ at room temperature. The greater surface area and presence of mesopores offered an increased number of adsorption sites for gas molecules, along with more sites for the reaction of adsorbed oxygen ions with formaldehyde. Notably, the sensor response towards formaldehyde increased linearly along with both light intensity and gas concentration [46,59].

Besides metal oxides, metal sulfides consist of another big group of photocatalyst materials [61]. In recent years, many research articles have been published focusing on the gas-sensing performances of metal sulfides under visible light illumination. Under green light illumination, the SnS_2_-based gas sensor exhibits superior sensing properties towards NO_2_, with a limit of detection of 38 ppb, while offering complete recovery at room temperature. Under illumination, the higher electron population in the conduction band of SnS_2_ attracted a greater number of NO_2_ molecules, resulting in an enhancement in sensitivity. Furthermore, the gas response and the response/recovery time of the sensor decreased from 10.8% to 5.7% and from 164/236 s to 114/155 s as the intensity of the green light increased from 187 mW cm^−2^ to 637 mW cm^−2^, respectively. According to the authors, it may be due to the fact that the green light generates more electron–hole pairs and also reduces the amount of adsorbed oxygen ions on the surface of the sensing material. It also indicates that the photon energy of the green light wavelength increases the adsorption and desorption rates of the NO_2_ on the surface of the sensing material [62]. Exactly the same effect was observed on CdS nanoflake-based sensors under green light illumination at room temperature. When the light intensity increased from 0.25 mW cm^−2^ to 2.1 mW cm^−2^, the response/recovery time and the relative response of the sensor towards 5 ppm of NO_2_ gas concentration decreased from 47/1860 s to 44/113 s and from 182% to 89%, respectively (Table 1). More interestingly, under fluorescent lamp (0.25 mW cm^−2^) and natural solar light (approx. 110 mW cm^−2^), CdS nanoflake-based sensor exhibits sensing response of 54% and 57% towards 1 and 20 ppm of NO_2_ with response/recovery time of 146/670 s and 13/123 s, respectively. According to the authors, the superior gas sensing properties of the sensor are attributed to the morphology of the nanoflake array, notably to its minimal band gap energy (2.4 eV). These inherent features of CdS significantly enhance both light absorption and conductivity [63].

Z. Lin et al. focused on employing cadmium selenide (CdSe) nanoribbons in the fabrication of visible light-enhanced ethanol gas sensors [64]. They utilised a 350 W xenon lamp as an external source to ramp up the sensor ability. As the same phenomenon discussed earlier, the gas response of CdSe nanoribbon-based sensors towards 50 ppm of ethanol at 200 °C slightly increased, approximately from 1.0% to 1.1% with increasing light intensity from 5.92 mW to 12.18 mW, respectively. Interestingly, forming ternary compounds (CdS_x_Se_1−x_) using those two semiconductor materials discussed above (CdS and CdSe) facilitates wavelength-tuneable light adsorption in the visible region due to their continuously tuned direct band gap from 1.72 to 2.42 eV [65]. The optimum operating temperature of CdS_x_Se_1−x_ nanoribbon-based gas sensors, in the dark, towards acetic acid, is 200 °C and it has no effect towards target gas below 160 °C. As shown in Figure 3, at higher temperatures, the RR of the sensor towards 2 ppm acetic acid is approximately 0.4% with a Limit of Detection (LOD) of 1.13 ppm. However, under visible light illumination (12.18 mW Xe lamp), the optimum temperature could be divided by two (100 °C), and the RR value (ca. 0.33% at 2 ppm) is almost as good as this at 200 °C in the dark (Table 2). Visible light illumination helps to obtain the lowest LOD of 0.87 ppm at 100 °C. Moreover, under visible light illumination, even at room temperature, the CdS_x_Se_1−x_ nanoribbon-based gas sensor shows a promising sensing response, ca. 0.18% at the same concentration, with an LOD of 1.03 ppm.

In the last two decades, apart from inorganic materials, the sensing efficiency and stability of organic semiconducting materials has gathered lots of attention from researchers [66]. However, the light effect on molecular semiconducting materials only started to be used in gas-sensing applications very recently. Thus, M. Elakia et al. studied gas sensors based on carbon nanotubes coated with pyrene, using white light as an external trigger to activate and enhance the sensing performances of the sensors [67]. The contact potential difference in both pristine and pyrene-derivatized (pyrene tetra topic ligand: PTL) multi-walled carbon nanotubes (MWCNTs), determined by scanning Kelvin probe system, remains unaffected, and the response is hardly detectable when exposed to volatile organic compounds in a dark environment (Figure 4). Notably, the sensor is activated under visible light illumination and exhibits increasing in resistance during exposure to triethylamine (TEA). The authors explained this behaviour by considering that, in the dark, the highest occupied molecular orbital (HOMO) of pyrene is completely occupied, preventing electron donation from the amine. In contrast, this HOMO, partially depleted under light, can readily accept electrons from the adsorbed TEA molecule. Finally, the injection of these electrons into the MWCNTs results in enhancement in sensing performance, its p-type nature leading to a resistance increase.

### 3.2. Heterojunction Gas Sensors

Different gas-sensitive materials engaged in heterostructures were continuously reported to offer better gas-sensing properties, in terms of sensing response, selectivity, operating temperature and reproducibility, compared to chemoresistors prepared with the same sensing materials [68,69]. However, the interaction between sensing materials and gas molecules is identical in heterojunction devices as in chemoresistors, but the sensing mechanisms of heterojunction devices are more complex. Indeed, after adsorption, charge transfer between target gas and sensing material can lead to doping or trapping, and more important, it can modify the energy barrier at the interface between materials, which gives us astonishing results.

Numerous studies have been carried out on ZnO engaged in a heterostructure with other metal oxides, metal sulphides, quantum dots and noble metals [70,71,72]. Thanks to their difference in work function, each combination gives rise to different interfaces with different levels of energy barrier, which play a major role in their optical, electrical and gas sensing properties. Most interestingly, engaging the ZnO, which has a wide band gap of 3.37 eV in a heterostructure combined with photosensitive or low band gap materials, expands the photo-response of ZnO to the visible region. Moreover, these properties can be modified by modifying the thickness or, more generally, the deposited quantity of sensing materials. Some noteworthy examples are summarised hereafter.

Combining oxygen vacancy (OV)-enriched ZnO nanorods with noble metal nanoparticles like Pd in hybrid devices exhibits superior sensing performance under visible light illumination compared to ZnO chemoresistor [73]. To achieve oxygen vacancy enrichment, ZnO nanorods underwent treatment with H_2_ (10% in Ar), and subsequently, Pd nanoparticles were dispersed via a solvent reduction method. The presence of oxygen vacancies was confirmed via X-ray photoelectron spectroscopy analysis. Figure 5 shows the dynamic response curves of each fabricated gas sensor towards 0.1% CH_4_ exposure in the dark and under 590 nm light-illuminated conditions (6 mW cm^−2^).

With and without oxygen vacancy enrichment, ZnO nanorods (chemoresistor) displayed no gas response to CH_4_ under both dark and visible light conditions [73]. This observation suggested that oxygen vacancies alone do not contribute to light-activated CH_4_ sensing at low temperatures. Additionally, it is important to mention that CH_4_ is a chemically inert gas at room temperature, making it challenging to detect [74]. However, taking advantage of the heterostructure, the sensors based on the ZnO/Pd and OV ZnO/Pd exhibit a slight increase in relative response of 1.1% and 5.4% towards 0.1% of CH_4_ gas, with a response/recovery time of 7/5 min in the dark, respectively. This increase in response to heterostructure devices is due to the catalytic effect of Pd nanoparticles. Most interestingly, under visible light illumination, the RR value of ZnO/Pd was multiplied by two (2.3%), while this of the OV ZnO/Pd-based sensor was multiplied almost by six (36.8%), and the response/recovery time was reduced to 4/4.5 min, respectively. The OV-enriched ZnO/Pd hybrids show dramatic enhancement in the gas response under light illumination compared to dark conditions, demonstrating that the combined effects of oxygen vacancies and Pd nanoparticles played a crucial role in light-activated CH_4_ sensing. Figure 6a,b illustrate the dynamic relative response curves and fitted responses of the oxygen vacancy-enriched ZnO/Pd hybrid gas sensors to varying concentrations of CH_4_ (0.01–1%) under visible light illumination. When exposed to different target gases (CH_4_, CO and H_2_S), oxygen vacancy-enriched ZnO/Pd hybrids exhibited highly selective detection of CH_4_ under visible light illumination compared to CO and H_2_S (Figure 6c).

Upon contact between Pd and ZnO, a Schottky junction forms at ZnO/Pd hetero-interfaces due to the difference in their work functions. It makes it difficult for electron transfer in the absence of surface oxygen vacancies. The introduction of oxygen vacancies in ZnO lowers the energy barrier, facilitating the direct transfer of electrons from Pd to ZnO. Surface oxygen vacancies on ZnO act as active centres and play a key role in promoting photocatalytic CH_4_ oxidation. Hence, the enhanced CH_4_ sensing in the OV ZnO/Pd sensor under visible light is attributed to photo-excited electrons from Pd promoting highly active oxygen species formation on ZnO that facilitates the CH_4_ oxidation reaction, resulting in a superior sensing response under visible light illumination.

A similar photocatalytic effect was also observed in In_2_O_3_-sensitised ZnO nanoflower-based formaldehyde (HCHO) gas sensors. The ZnO nanoflowers were synthesised by a simple co-precipitation method followed by a hydrothermal treatment. The as-prepared product was treated with In(NO_3_)_3_ aqueous solution once, twice and thrice and heated to form different levels of In_2_O_3_-sensitised ZnO nanoflowers ZNFI-1, ZNFI-2 and ZNFI-3, respectively. Due to the different work functions between ZnO and In_2_O_3_, a heterostructure is formed with a depleted zone at the interface between ZnO and In_2_O_3_. The photocurrent intensity of the ZnO nanoflower-based gas sensor gradually increased with the increasing concentration of the sensitiser (In_2_O_3_). Most interestingly, a higher concentration of the sensitiser expands the photocurrent response of the ZnO nanomaterials to the visible region. Upon exposure to light with a wavelength of 460 nm (0.213 mW cm^−2^), the flowerlike ZnO samples with an In_2_O_3_ coating on their surface exhibit enhanced absorption of visible light, leading to the generation of photo-excited electrons. The migration of these electrons from the bottom of the conducting band of In_2_O_3_ to the conducting band of ZnO, avoiding the recombination of photo-excited holes and electrons in In_2_O_3_-sensitized ZnO-based sensors. This effect improves the efficiency in the separation of charge carriers. Moreover, an increase in the quantity of In_2_O_3_ nanoparticles correlates with a higher injection of photo-excited electrons into ZnO. When HCHO was injected into the gas chamber, the sample that was treated twice with In(NO_3_)_3_ aqueous solution exhibited the highest gas response (419%) at 100 ppm of HCHO concentration, which is 5 times larger than that of ZnO nanoflower chemoresistor and 1.6 times larger than other two devices at room temperature. These results show that the sensitivity of the sensor towards target gas depends not only on the generation and transport of photo-generated charge carriers but also on the surface properties of the sensing material. Indeed, increasing the concentration of the sensitiser (In_2_O_3_) increases the photocurrent intensity, but, due to the serious agglomeration of In_2_O_3_ nanoparticles at higher concentrations of sensitiser, leads to a decrease in the active sites on the surface of ZnO, resulting in the decrease in the gas response of the ZNFI-3 sensor [75].

B. Wu et al. fabricated ZnO/CdSe heterostructure sensors by depositing ZnO nanorods onto the surface of CdSe nanoribbons [76]. When the ZnO sensor is exposed to ethanol, at the interface of the heterostructure, the reaction with chemisorbed oxygen species, $O^{-}$ and $O^{2-}$, releases electrons that move from the surface back into the conduction band of ZnO, resulting in a current increase (Equation (3)).
(3)$$CH_{3}CH_{2}OH\;\left(ads\right)+O^{2-}\left(ads\right)\to C_{2}H_{4}O+H_{2}O+2e^{-}$$

In ZnO/CdSe heterostructure, this phenomenon is easier. Simultaneously, generated holes in CdSe are neutralised by the reaction of oxygen anions, $O^{-}$ and $O^{2-}$, which induces more active sites on the sensing material, resulting in a gas response of 2.4% in the dark at 160 °C, which is 20 times higher compared to ZnO chemoresistor in the dark at its optimum temperature (260 °C). Upon visible light illumination (Xe lamp, 12.8 mW), CdSe nanoribbons generate electron–hole pairs due to their weak band gap (1.7 eV), which do not efficiently recombine. Instead, the electrons are transferred from CdSe to ZnO, accumulating at the lower conduction band of ZnO (Figure 7). Recombination of holes and adsorbed anionic oxygen species gives adsorbed O_2_ molecules, which can then be desorbed as the form of O_2(g)_, as depicted in Equation (1). This accumulation results in an increased conductance in the ZnO/CdSe heterostructure. Consequently, the response of the ZnO/CdSe-based gas sensor at 25 ppm of ethanol vapour increased to ca. 7.5%, which is 3.6 times higher compared to the response in the dark. A schematic energy level model for ZnO/CdSe-based heterostructure sensors when exposed to ethanol is shown in Figure 7.

Apart from nanorods and nanocomposites, the utilisation of quantum dots (QDs) to improve the sensing capability also gathered lots of attention. A. Chizhov et al. prepared hybrid materials based on nanocrystalline ZnO sensitised with two types of core/shell nanoparticles with different localisations of photo-excited electron–hole pairs between the core and the shell [77]. They demonstrated the formation of ZnO/CdS(shell)@CdSe(core) and ZnO/ZnSe(shell)@CdS(core) heterostructures with a straddling gap and a staggered gap arrangement, respectively. In this work, they use visible green light (535 nm) to enhance the sensing properties of the sensors in the presence of 0.12–2 ppm NO_2_ in air. Under illumination, in ZnO/CdS(shell)@CdSe(core) arrangement, photo-generated holes are confined within the CdSe core, effectively passivating the CdSe core defect states that increase the transfer of photo-generated electrons to the conduction band of ZnO. These charges recombine with NO_2_ molecules during the dark interval of the measurement cycle. This leads to increased sensitivity to NO_2_ in ZnO/(CdS@CdSe) heterostructure. In the ZnO/ZnSe(shell)@CdS(core) arrangement, the shell formation similarly passivates CdS core defect states. However, in this case, photo-generated holes are localised in the ZnSe shell, facilitating efficient charge transfer from chemisorbed $NO_{2}^{-}$_(ads)_ and their subsequent desorption under illumination. This unique combination of factors in ZnO/ZnSe@CdS heterostructure results in a substantial increase in sensitivity to NO_2_ under visible light at room temperature compared to ZnO/(CdS@CdSe) heterostructure device in the studied range (0.12–2 ppm).

The same authors also studied a series of metal oxides like ZnO, SnO_2_ and In_2_O_3_ sensitised with CdSe quantum dots (QD) [35]. Compared to pristine forms, sensitising these metal oxides with CdSe QDs leads to a higher response to NO_2_ exposure under illumination of green light (535 nm). Upon visible light illumination, electron–hole pairs generated in CdSe QDs facilitate the transfer of photo-excited electrons to the conduction band of the metal oxide semiconductor matrix, while the photo-excited holes can recombine with electrons that are trapped by the chemisorbed $NO_{2}^{-}$_(ads)_. Due to the injection of $e^{-}$ from the CdSe QDs, the density of charge carriers in metal oxide increases, resulting in a higher sensing response under visible light illumination. They also found that the most effective sensitisation was achieved for In_2_O_3_ because of the maximum band offset between In_2_O_3_ and CdSe QDs.

To date, many research articles explored the combination of carbon nanomaterials like carbon nanotubes or graphene derivatives with metal oxides, metal sulfides and metal nanoparticles. Thus, Pd-decorated ZnO/rGO [78], SnS_2_/rGO [79], MoS_2_/rGO [80], Ag nanoparticle-decorated on rGO [81], Ti/graphene and WO_3_/GO-based devices [82,83] were studied as gas sensors under visible light illumination, towards different oxidative and reductive gases. This wide range of studies gives us in-depth knowledge about the gas-sensing properties of rGO as engaged in heterostructures with different notable gas-sensing materials. Some interesting results are discussed below.

Under visible light illumination (460 nm), hydrothermally fabricated MoS_2_ nanosheets on rGO nanosheets-based heterostructure sensors show a dramatic enhancement in gas sensing properties in terms of gas adsorption–desorption kinetics and sensing response. Upon exposure to 10 ppm of HCHO gas under visible light illumination, MoS_2_/rGO-based heterostructure sensors exhibit a sensing response of 64% with a fast response time of 17 s, which is over 8 times higher than the response of the heterostructure sensor in the dark (8.5%), at room temperature [80]. As per the authors, this enhancement was not solely attributed to increased photoconductivity. Their studies proposed that photocatalytic oxidation of HCHO on the MoS_2_ surface played a crucial role in the improved sensing performance under visible light illumination. Surface oxygen species on MoS_2_ were investigated using XPS and O_2_-TPD (temperature-programmed desorption), revealing an increase in chemisorbed oxygen species upon light exposure. In situ IR spectra confirmed the photocatalytic oxidation of HCHO to CO_2_ and H_2_O on the sensing layer under visible light, which was also observed by L. Han et al. [75]. These results suggested that the light-activated HCHO sensing behaviour of MoS_2_/rGO-based hybrid sensor is directly linked to their photocatalytic activity, similar to the phenomenon we discussed earlier with oxygen vacancy-enriched ZnO/Pd-based hybrid sensor studied by R. Chen et al. [73]. Additionally, the authors compared the efficiency of MoS_2_/rGO-based heterostructure sensors with MoS_2_ and rGO chemoresistors under the same experimental conditions. MoS_2_ chemoresistor exhibits a gas response of 2.5% in the dark and 11.5% under light, while the response of rGO chemoresistor remains almost the same (5.4%) in the dark and (5.6%) under visible light illumination. Engaging MoS_2_ with rGO in heterostructure significantly improves the sensing properties, both in the dark and under visible light illumination, compared to their chemoresistors. This observation also portrays the advantages of heterostructure devices over chemoresistors.

Similarly, as an n-type semiconductor, SnS_2_-based chemoresistor shows no acceptable response towards low concentrations of NO_2_ at room temperature. Introducing a minimal amount of rGO (1 mg) with SnS_2_ induces dramatic enhancement in relative response in the range of 0.125–1 ppm, with a quick response/recovery time of 75/242 s at room temperature [79]. Increasing the concentration of rGO by a factor of 20 in SnS_2_-based heterostructure devices leads to inversion in the polarity of the device (n-type to p-type), resulting in inversion of the sensing response, higher drift and decreasing in gas response. To investigate the effect of visible light on gas sensing properties of SnS_2_/rGO-based nanohybrid gas sensors, varying light wavelengths (blue, green, and red) and intensities were employed during NO_2_ exposure. The p-type SnS_2_/rGO nanohybrids showed negligible improvement under light due to rGO’s limited photoelectric properties, as observed earlier [80]. However, the sensing properties of the n-type SnS_2_/rGO heterostructure sensor enhanced under visible light illumination, which allowed for the detection of lower NO_2_ concentrations, down to 10 ppb of NO_2_. The authors analysed the amplification factor (RR_light_/RR_dark_) and relative response/recovery time (t_res light_/t_res dark_ and t_rec light_/t_rec dark_) of SnS_2_/rGO heterostructure gas sensors. Interestingly, they found that the highest amplification factor (ca. 5.9) was observed for the lowest intensity (650 nm, 1 mW cm^−2^), revealing an optimal light intensity for the gas sensing reaction (Figure 8a,b). Meanwhile, recovery time is reduced under low-power illumination by a factor of 2, while response time remains quasi-unchanged. The proposed band alignment diagram at the heterojunction of SnS_2_/rGO nanohybrids is shown in Figure 8.

These results also demonstrate that the generated electron–hole pairs do not interact proportionally with the light intensity with absorbed species. As a result, higher light intensity shifts the balance towards the desorption rate rather than the adsorption process. A similar effect was observed by M. Zhao et al. when they tested Ti/graphene-based heterostructure gas sensors under high-power light illumination [82]. They developed a photoactivated NH_3_ gas sensor working at room temperature by depositing varying thicknesses of Ti onto the graphene surface. The optimised configuration involved a 5 nm thin film of Ti, which was followed by its oxidation and the formation of titanium oxide with different Ti valencies, the lower valencies leading to reduced optical bandgap. Similar to the previous discussion, solely graphene-based chemoresistor exhibits a very low RR of about 0.2%, towards 400 ppm of NH_3_ concentration both in the dark and under visible light illumination. As expected, when graphene is engaged with Ti in a heterostructure device, it dramatically increases the RR to 7.5% in the dark and 17.9% under visible light illumination (0.75 W white LED), with a response/recovery time of 2.5/2.7 min, respectively, at room temperature. The resulting photo-excited electron–hole pairs, coupled with catalytic effects of TiO_x_/graphene, collectively contributed to enhancing the sensor’s response to NH_3_ gas.

X. Geng et al. studied the NO_2_ sensing properties of WO_3_/GO-based heterostructures under different visible light illuminations [83]. As discussed earlier, the electrical resistance of WO_3_/GO sensors decreases with increasing in light energy. Also, the response (R_NO2_/R_air_) of the sensor increases with increasing in light energy from 4.4 to 82.5 when wavelength varies from 640 nm to 530 nm, respectively (Figure 9, Table 3). However, excessively high photoenergy (530 nm–400 nm) results in decreases in response from 82.5 to 44.4. The generation of more electron–hole pairs on the surface as photon energy increases not only induces a decrease in the resistance of devices but also favours gas desorption, as above mentioned.

Notably, WO_3_/GO-based heterostructure sensors exhibit remarkably high responses of 82.5%, 63.8% and 44.4% to 0.9 ppm NO₂ under green, blue and purple light, respectively. The highest relative response was found under green light (530 nm) with a response/recovery time of 23.8/31.7 min, revealing the optimum light wavelength. However, the compromise between response and recovery times favours blue light, offering optimal conditions for WO_3_/GO sensors in shorting response/recovery time to 18.6/23.3 min, respectively, while maintaining an acceptable sensor response of 63.8%.

Numerous research works have delved into the properties of inorganic materials under light illumination in gas sensing applications, but very few scientific articles explored the effect of incorporation of molecular semiconductors in heterostructure devices under visible light illumination. One of the major benefits of molecular semiconductor-based devices is their attractive optical and electrical properties, including potential low operating temperature and comparatively high mechanical flexibility [84]. Additionally, their hydrophilicity/hydrophobicity and their p- or n-type nature can be tuned by molecular engineering in relation to the shift of their HOMO and LUMOs orbitals.

Perylenediimide (PI) is one of the most important classes of dyes and is intensively explored in the field of functional organic materials, especially for its outstanding optoelectronic properties [85]. When such material is engaged with SnO_2_ in a heterostructure, it increases the absorbance of the PI/SnSO_2_-based device. Despite having a wide band gap, SnO_2_ is significantly influenced by intense absorption bands (450 and 600 nm) of PI. The sensor shows a gas response of ca. 30% towards 0.5 ppm of NO_2_ in the dark at room temperature. As expected, when the sensor was illuminated by white LED light with a wavelength between 400 and 700 nm, it exhibited astonishing enhancement in the sensing response with an RR of 131.6%, which is more than 4 times higher compared to this in the dark, with response/recovery time of 6/4 min [31]. The authors explained the enhancement of the sensing properties of the heterostructure device using similar photo-excited electron–hole pair generation and electron transfer phenomenon to the inorganic semiconductor, which we discussed before.

## 4. Light Effect on Ambipolar Organic Electronic Devices

As we said earlier, light illumination on heterojunction can modify the density of charge carriers of the sensing layer. However, the effect of light also depends on interfacial charge accumulation and on the work function of the materials engaged in heterojunction. A few years ago, we demonstrated a brand-new effect on gas sensors using visible light illumination on organic heterojunction devices [18].

We fabricated a series of bilayer semiconductor devices consisting of copper phthalocyanines with different degrees of halogenation, such as CuF_16_Pc, CuF_8_Pc, CuCl_8_Pc and CuPc as sublayers (Figure 10a), which were commonly covered by a highly conducting molecular material, lutetium bis-phthalocyanine (LuPc_2_) as a top layer (Figure 10b) [86]. This type of heterojunction device was initially reported and patented by one of us [87]. The electrical and NH_3_ gas sensing properties of the organic heterojunction devices were investigated both in the dark and under visible light illumination (red LED, 642 nm) at room temperature.

The CuF_16_Pc/LuPc_2_ device (Figure 11c) exhibits a positive response to NH_3_, whereas the CuCl_8_Pc/LuPc_2_ (Figure 11a) device exhibits a negative response, the same as for CuPc/LuPc_2_ [88]. All three of these devices exhibit excellent observable response towards NH_3_ gas at room temperature in the dark, and the response remains almost the same under visible light illumination, while a current increase was observed due to light absorption. Unlike previously discussed sensors, we did not observe any enhancement in gas sensing properties upon illumination for the octa-fluorinated copper phthalocyanine-based heterojunction device (CuF_8_Pc/LuPc_2_) (Figure 11b), but the most interesting is the inversion in the nature of the response. So, apart from charge density modification, it was shown for the first time that light could also induce a new type of phenomenon in gas sensors called the detrapping effect (Figure 10c) [18], which can inverse the polarity of majority charge carriers. Initially, the CuF_8_Pc/LuPc_2_-based heterojunction device shows a very clear negative response (with less noise) towards 20 ppm of NH_3_ gas with RR of −7%, displaying its p-type nature in the dark, but under visible light illumination, the device exhibits stable positive response with RR ca. 1.2% towards the same concentration of NH_3_ gas showing its n-type nature. Therefore, the CuF_8_Pc/LuPc_2_-based device changes its polarity from p- to n-type, depending on the optical external trigger, exhibiting its bistability.

This fascinating phenomenon under visible light illumination can be explained by the desorption effect of oxygen molecules. It is well known that the adsorption of oxygen molecules on semiconductor devices can decrease the density of free electrons while increasing hole density [89]. At the same time, light illumination on semiconducting devices is well known for facilitating oxygen desorption from the surface of sensing materials, which induces a detrapping of negative charges (Figure 10c). Thus CuF_8_Pc/LuPc_2_-based device exhibits p-type behaviour towards reduction gas (NH_3_) in oxygen-rich dark environmental conduction. However, under red light illumination, the desorption of oxygen molecules is favoured. Hence, there is an inversion in nature of majority charge carriers, which leads to an inversion in gas response. This inversion is mainly due to the ambipolar property of CuF_8_Pc, with an initial density of positive and negative charge carriers very near to each other [90]. Since the electrical properties of the semiconductor material are near equilibrium, a weak energy coming from an external trigger is sufficient to inverse its polarity.

To date, this is the only report of such an astonishing effect in conductometric sensors using visible light. This type of ambipolar device can replace two different sensors into one, which can help us to reduce the size of the sensing system and the cost of fabrication. However, having control over majority charge carrier inversion needs in-depth molecular engineering to shift the energy levels near equilibrium and mastering external triggers still remain challenging. Furthermore, more studies are needed before pushing ambipolar gas sensors into real-life applications.

## 5. Conclusions

In this article, we reviewed the effect of visible light illumination on inorganic and organic semiconductor-based chemoresistors and heterojunction gas sensors. The first involved phenomenon is the photo-generation of additional charge carriers that induce a current increase flowing through the electronic devices. Then, generated holes can facilitate the desorption of molecular oxygen, but in some cases, photo-generated electrons increase their adsorption. Photo-generated holes and electrons can also react with the target gas, depending on its nucleophilicity and electrophilicity. Compared to chemoresistors, heterostructures offer additional opportunities; one of the materials absorbs light, and then electron transfer occurs towards the second material. The interest of visible light illumination is to improve the sensing performances by reducing the operating temperature, reducing response time, increasing sensitivity and, in some cases, improving selectivity. Obviously, light effects depend on light energy and power. Finally, we also report a particular visible light effect with an ambipolar heterojunction device that is the inversion in majority charge carriers, allowing the device to operate in two different states. Clearly, illumination is a wonderful tool that should be used more in conductometric gas sensors. However, wavelength and power light have to be optimised for each sensing material and transducer. In the future, integration of a light source in sensing systems should be as common as heating systems currently implemented in commercially available gas sensors.

## Figures and Tables

**Figure 1 sensors-24-01571-f001:**
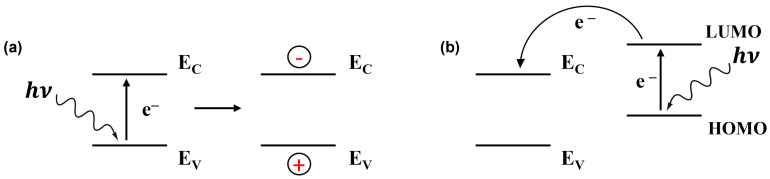
Schematic view of light-induced charge carrier generation in an inorganic semiconductor (E_V_ and E_C_ are the top of the valence band and the bottom of the conduction band, respectively) (**a**) and in a dye/inorganic semiconductor heterojunction device (HOMO and LUMO are the highest occupied and lowest unoccupied molecular orbitals, respectively) (**b**).

**Figure 2 sensors-24-01571-f002:**
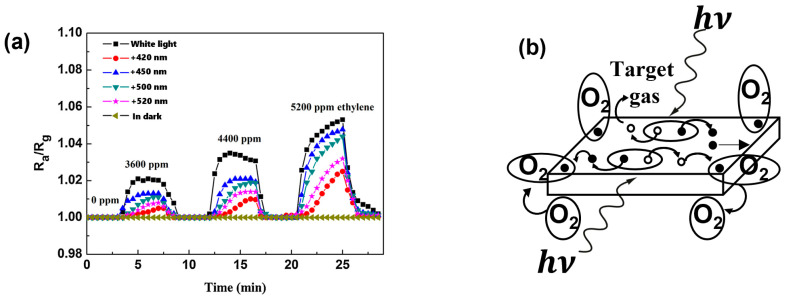
Sensing responses of a ZnO sensor to different concentrations of 
ethylene in the air as a function of time in the dark or under visible light 
irradiations (**a**) and diagram illustrating the visible light-activated 
gas sensing mechanism of ZnO sensor at room temperature showing the reaction of 
O_2_ with electrons (
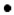
) and of target gas with holes (
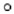
), both $e^{-}$ and $h^{+}$ being photo-generated (**b**) (modified from [47]).

**Figure 3 sensors-24-01571-f003:**
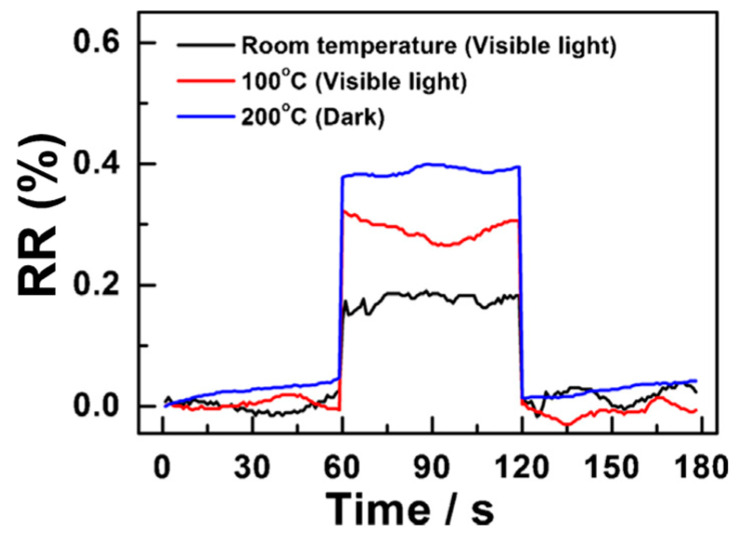
Relative response of CdS_x_Se_1−x_ nanoribbon-based sensors upon exposure to 2 ppm CH_3_COOH under different conditions (adapted from [65]).

**Figure 4 sensors-24-01571-f004:**
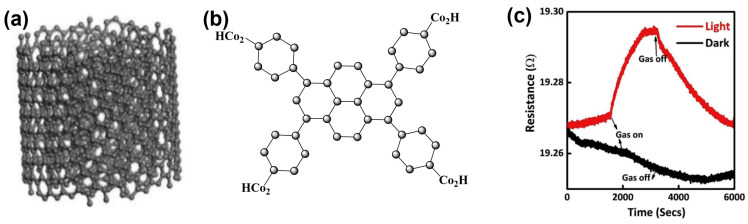
Molecular structures of MWCNTs (**a**) and PTL (**b**). Typical sensor response of PTL functionalised MWCNTs with 20% TEA exposure under dark and light conditions (**c**) (modified from [67]).

**Figure 5 sensors-24-01571-f005:**
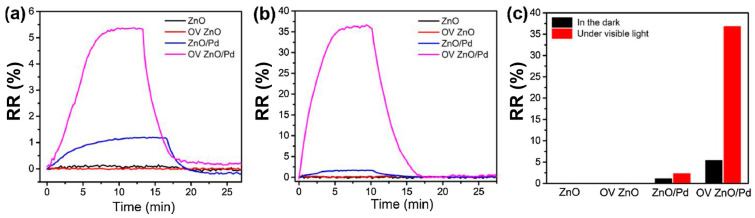
Dynamic response curves of the sensors based on ZnO, OV ZnO, ZnO/Pd and OV ZnO/Pd samples to 0.1% CH_4_ in the dark (**a**) and under 590 nm light illumination (6 mW cm^−2^) (**b**); (**c**) the corresponding gas sensing responses (adapted from [73]).

**Figure 6 sensors-24-01571-f006:**
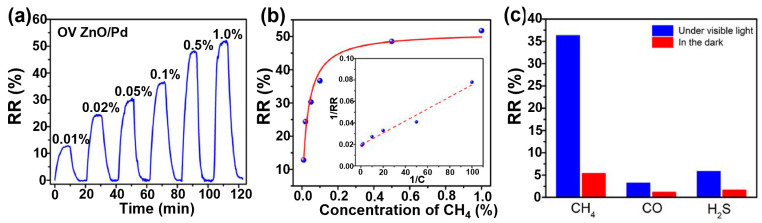
Dynamic response curve (**a**) and fitted responses (**b**) of the OV ZnO/Pd sensor to 0.01–1% CH_4_ under 590 nm light illumination (6 mW cm^−2^), the inset confirms the Langmuir’s law; (**c**) selectivity of the OV ZnO/Pd sensor to 0.1% CH_4_ over CO and H_2_S under 590 nm light illumination (6 mW cm^−2^) (adapted from [73]).

**Figure 7 sensors-24-01571-f007:**
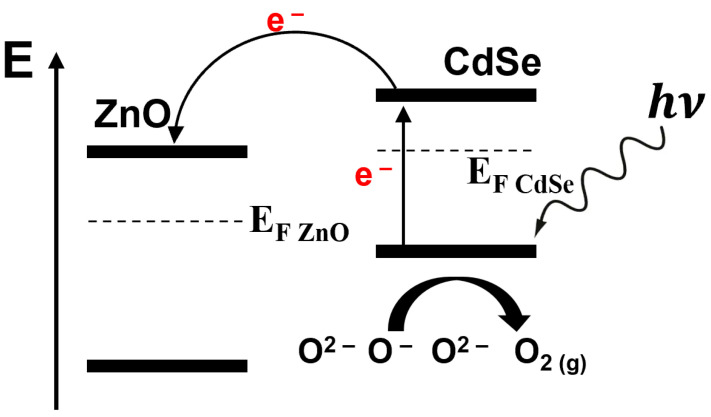
Schematic model for ZnO/CdSe heterostructure-based sensors when exposed to ethanol. E_F_ indicate Fermi levels of both materials (modified from [76]).

**Figure 8 sensors-24-01571-f008:**
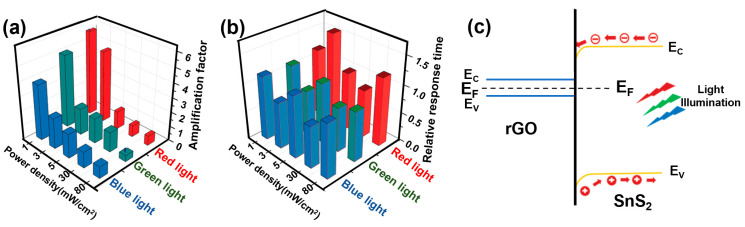
The amplification factor of sensitivity (**a**) and relative response time (**b**) under light illumination of different wavelengths and power densities. The band alignment diagram of the light-assisted SnS_2_/rGO sensors (**c**). The light-induced electron–hole pairs are generated and separated in the conduction and valence bands of SnS_2_ (adapted from [79]).

**Figure 9 sensors-24-01571-f009:**
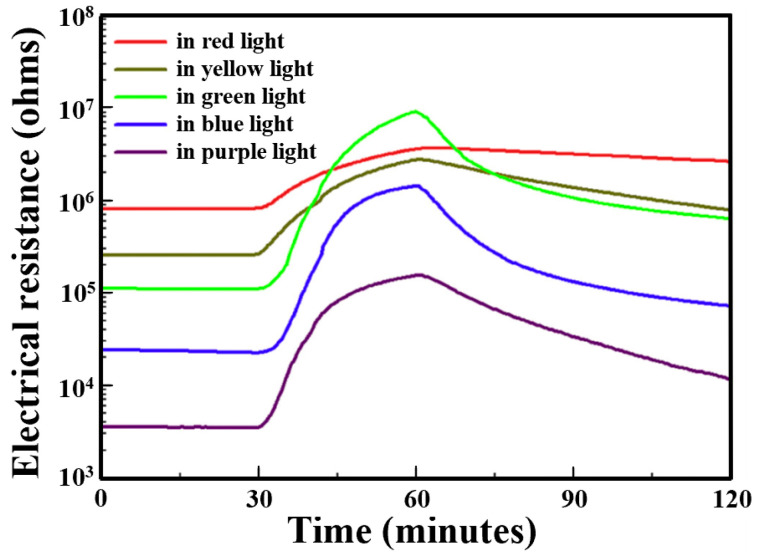
Electrical resistance responses of the WO_3_-GO sensors to 0.9 ppm NO_2_ gas performed under light illumination at different wavelengths at room temperature (adapted from [83]).

**Figure 10 sensors-24-01571-f010:**
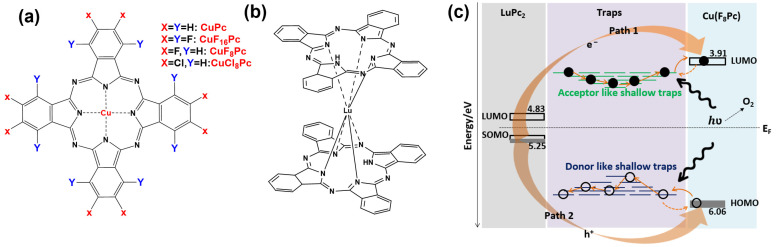
Chemical structure of copper phthalocyanine with different degrees of halogenation (**a**) and lutetium bis-phthalocyanine (LuPc_2_) (**b**). The charge transport scheme between frontier orbitals and trap states in Cu(F_8_Pc) and LuPc_2_ under visible light exposure (**c**) (modified from [18]).

**Figure 11 sensors-24-01571-f011:**
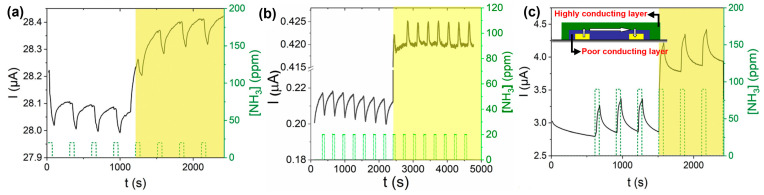
Response of CuCl_8_Pc/LuPc_2_- (**a**), CuF_8_Pc/LuPc_2_- (**b**) and CuF_16_Pc/LuPc_2_- (**c**) based heterojunction sensors towards 20 ppm of NH_3_ under dark and light illumination (right sides of each curve) at room temperature. The inset of (**c**) represents a simplified device scheme (adapted from [18]).

**Table 1 sensors-24-01571-t001:** Comparison of t_90_, recovery time and RR of CdS nanoribbon-based sensors under different intensities of green light (adapted from 63).

CdS Nanoribbon-Based Sensors	Green Light	
	0.25 mW cm^−2^	2.1 mW cm^−2^
t_90_ (s)	47	44
t_recov._ (s)	1860	113
RR (%)	182	89

**Table 2 sensors-24-01571-t002:** Comparison of RR and LOD of CdS_x_Se_1−x_ nanoribbon-based sensors under different conditions [65].

	Conditions	Dark	RT—Visible Light	200 °C—Dark
Parameter				
RR (%)		0	0.33	0.4
LOD (ppm)		-	0.87	1.13

**Table 3 sensors-24-01571-t003:** Sensing characteristics of the WO_3_-GO sensor to 0.9 ppm, NO_2_ illuminated at different wavelengths (adapted from [83]).

Wavelength (nm)	Light Colour	Sensor Response (R_NO2_/R_air_)	Response Time (min)	Recovery Time (min)
640	Red	4.41	28.7	>60
580	Yellow	10.66	26.2	>60
530	Green	82.47	23.8	31.7
480	Blue	63.73	18.6	23.3
400	Purple	44.34	20.3	54.6

## Data Availability

Not applicable.

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
