# Peer review of "Effects of Visible Light on Gas Sensors: From Inorganic Resistors to Molecular Material-Based Heterojunctions"

_sensors, 2024, doi:10.3390/s24051571_

Round 1

Reviewer 1 Report

Comments and Suggestions for Authors

In this manuscript (sensors-2876549), the authors reviewed the effects of visible light on gas sensors. The topic is interesting and can attract a wide range of readerships, but there are some problems in the introduction, presentation and discussions. As such, the following revisions are needed before possible acceptance. My specific comments are as follows:

1. Introduction: What is the background of light excited gas sensors? What are the advantages of light excited gas sensors? What problems can be solved of light excited gas sensors. For example, solving the problem of high operating temperature of oxide gas sensors. Generally, room temperature gas sensors have low sensitivity and slow response speed. Suggest combining the research progress of heating gas sensors, highlighting the current research status of visible light gas sensors, and may refer to Sens. Actuators B Chem. 2024, 402, 135136.

2. “4. Resistive gas sensors”. To distinguish from “5. Heterojunction gas sensors”, “single material based gas sensors” can be used. Or other expression.

3. As a review article, it involves citations of many images. The sensor parameter definitions in these images are not uniform. In this case, it is recommended to add definitions for gas sensor response, sensitivity, and response/recovery times.

4. Conclusions: Suggest proposing constructive suggestions from different perspectives.

5. The quality of the images is poor, such as Figure 9. Suggest selecting some high-quality images.

6. English writing needs further polishing.

Comments on the Quality of English Language

English writing needs further polishing.

Author Response

Dear reviewer,

Please find hereafter our response point by point to your comments. 

Reviewer 1

In this manuscript (sensors-2876549), the authors reviewed the effects of visible light on gas sensors. The topic is interesting and can attract a wide range of readerships, but there are some problems in the introduction, presentation and discussions.

R: We thank the reviewer for the positive comments.

As such, the following revisions are needed before possible acceptance. My specific comments are as follows:

  1. Introduction: What is the background of light excited gas sensors? What are the advantages of light excited gas sensors? What problems can be solved of light excited gas sensors. For example, solving the problem of high operating temperature of oxide gas sensors. Generally, room temperature gas sensors have low sensitivity and slow response speed. Suggest combining the research progress of heating gas sensors, highlighting the current research status of visible light gas sensors, and may refer to Sens. Actuators B Chem. 2024, 402, 135136.

R: We consider that we answered this question in the introduction. In particular, we indicated that light illumination allows operating at lower temperature, then limiting the power consumption. Regarding the general high operating temperature of metal oxide-based gas sensors, we talked about the historical work of T. Seiyama (page 3, L113). Then, we cite the work by Cheung on light-assisted gas sensors, light allowing operating sensors at room temperature (L121).

However, we modified the abstract to enlight these points: Most interestingly, the light-activated gas sensors show promising results, particularly using visible light as an external trigger that lowers the power consumption as well as improves the stability, sensitivity and safety of the sensors. It effectively eliminates the possible damage on sensing material caused by high operating temperature or high energy light.

About the second part of the comment, the reviewer asked us to talk about the thermal effect. It is not the topic of the present review. Actually, in one particular example, we described how visible light illumination leads to an equivalent relative response as this obtained by heating at 200°C (L261-266, and Table 2).

  1. “4. Resistive gas sensors”. To distinguish from “5. Heterojunction gas sensors”, “single material based gas sensors” can be used. Or other expression.

R: We thank the reviewer for this remark. This is true, resistive gas sensors is too general. We replaced the title of section 4 by “Chemoresistors”. Additionally, we checked all along the manuscript the use of “resistors” and “resistive sensors”.

  1. As a review article, it involves citations of many images. The sensor parameter definitions in these images are not uniform. In this case, it is recommended to add definitions for gas sensor response, sensitivity, and response/recovery times.

R: We thank the reviewer. We paid attention to this point, and added the definition of the response and relative response as soon as it seemed to us ambiguous in the initial version.

  1. Conclusions: Suggest proposing constructive suggestions from different perspectives.

R: We thank the reviewer for this important point. We added a few perspectives at the end of the conclusion: Clearly illumination is a wonderful tool that should be more used in conductometric gas sensors. However, wavelength and power light have to be optimized for each sensing material and transducer. In the future, integration of a light source in sensing systems should be as usual as heating systems currently implemented in commercially available gas sensors.

  1. The quality of the images is poor, such as Figure 9. Suggest selecting some high-quality images.

R: We improved the quality of Figures 9, but also of Fig. 5, 6, 8 and 10c as much as possible.

  1. English writing needs further polishing.

R: About English writing, we did a particular reading to polish it, notably with the help of the second reviewer.

Sincerely

Reviewer 2 Report

Comments and Suggestions for Authors

In the presented review, an attempt is made to communicate the processes of visible light-activated gas sensitivity of resistive gas sensors (chemoresistors, chemosensors). The materials from which chemoresistors are formed may contain one substance or two (or more) substances. In the latter case, heterojunctions are formed in the structure of materials. The manuscript contains examples of materials of inorganic and organic nature.

The authors of the manuscript made a generalization of the processes without strict mathematical formulas, which facilitates their understanding by the reader. However, the absence in the review of the characteristics of gas-sensitive materials (film thickness, concentration and mobility of charge carriers) does not allow for a more accurate assessment of the processes occurring during visible light-activated chemisorption of gas molecules.

In my opinion, there are important conceptual inaccuracies in the manuscript:

1. Gas sensors of the resistive type are commonly called chemoresistors or chemosensors. The authors refer to them in the manuscript as resistors, and at the end of the manuscript as conductometric sensors. This leads to a misunderstanding of the text. It seems to me that this must be fixed.

2. The main carriers have a constant and unchanging charge. An inversion of the type of conductivity of the gas-sensitive material may occur, but not an inversion of the charge carriers. The authors mistakenly talk about the inversion of charge carriers (L52, 600-602,605, 609). This must be fixed.

3. The introduction does not sufficiently say that photoactivation depends on the intensity of light. In addition, it is desirable to indicate the intensity of light in the manuscript in the same units of measurement (L211, 219, 234, 240, 243, 320, 347, 368, 486, Table 1).

Remarks:

L8-13. These considerations should be transferred to the Introduction. They are not appropriate in the Annotation.

L39. Gas sensors with a light source do not always have low power consumption, since the light source requires additional energy.

L52, 600-602,605, 609. The main carriers have a constant and unchanging charge. An inversion of the type of conductivity of a gas-sensitive material, but not of charge carriers, may occur.

L64. Here, photoexcitation of charge carriers, but not of a semiconductor, occurs.

L77-82. The authors should mention here the light intensity for the considered wavelengths.

L88-89, 93. Phrases are incorrect.

L95. The equation is incorrect.

L126-127. The title of the section inaccurately reflects its content.

Fig.2, a. It is necessary to decipher the designations of the abscissa axis.

L172. Misspell.

L176-177. Maybe it is necessary to write "with an increase in the concentration of oxygen vacancies"?

L182-184. It is necessary to give the reaction equation.

L211, 219, 234, 240, 243, 320, 347, 368, 486, Table.1. It is desirable to indicate the light intensity in the same units of measurement.

L260. the abbreviation has not been deciphered.

L281. "Kelvin" is capitalized.

L279-284. It is necessary to explain what Kelvin probe force microscopy (KPFM) has to do with the gas sensitivity of the material.

L285. This is a repetition of the phrase mentioned above.

L315. What kind of resistor do the authors mean?

L2, 19,49,140,186, 315 and oth. Instead of resistors, you need to write chemoresistors.

L317-321. XPS Peaks of oxygen vacancies are not shown in Fig.5.

L327. The characterization of CH4 as an explosive gas is superfluous here.

L353. The expression in parentheses must be given in the form of an equation.

L404, fig.7. What happens to oxygen in the figure? This is not explained anywhere.

L428. The articles [81] and [38] have different authors.

L449,582, 599.What is "light illumination"?

L 462-464. The phrase is incomprehensible. What kind of resistor do the authors mean?

L572-579. Incorrect references to figures are mistakenly indicated in the text.

L573. There is a typo in the name of the heterojunction.

L600. What kind of molecule is "dioxygen molecules"?

L600-602, 609. The main carriers have a constant and unchanging charge. An inversion of the type of conductivity of the gas-sensitive material may occur, but not an inversion of the charge carriers. The authors mistakenly talk about the inversion of charge carriers. This must be fixed.

L605, 615. What type of sensors do the authors write about in the manuscript? About resistors, chemosensors, conductometric sensors?

Author Response

Dear reviewer,

Please find hereafter our response point by point to your comments.

In the presented review, an attempt is made to communicate the processes of visible light-activated gas sensitivity of resistive gas sensors (chemoresistors, chemosensors). The materials from which chemoresistors are formed may contain one substance or two (or more) substances. In the latter case, heterojunctions are formed in the structure of materials. The manuscript contains examples of materials of inorganic and organic nature.

The authors of the manuscript made a generalization of the processes without strict mathematical formulas, which facilitates their understanding by the reader. However, the absence in the review of the characteristics of gas-sensitive materials (film thickness, concentration and mobility of charge carriers) does not allow for a more accurate assessment of the processes occurring during visible light-activated chemisorption of gas molecules.

R: We thank the reviewer for his positive comments.

About the absence of characteristics of sensing materials, we feel that if we want to stay focus on light effect on gas sensing, we cannot dilute the text with structural, morphological or electrical properties of materials as mobility and density of charge carriers. So, it is the choice that we did. We consider that, if a sensing property is interesting for the reader, he can get all the information from the cited sources.

In my opinion, there are important conceptual inaccuracies in the manuscript:

  1. Gas sensors of the resistive type are commonly called chemoresistors or chemosensors. The authors refer to them in the manuscript as resistors, and at the end of the manuscript as conductometric sensors. This leads to a misunderstanding of the text. It seems to me that this must be fixed.

R: As answered to the reviewer 1 (point 2), we replaced the title of section 4 by “Chemoresistors”. Additionally, we checked all along the manuscript the use of “resistors” and “resistive sensors”.

  1. The main carriers have a constant and unchanging charge. An inversion of the type of conductivity of the gas-sensitive material may occur, but not an inversion of the charge carriers. The authors mistakenly talk about the inversion of charge carriers (L52, 600-602,605, 609). This must be fixed.

R: We thank the reviewer. We replaced “the inversion of charge carriers” by “the inversion of the nature of majority charge carriers”. With ambipolar materials or devices, we can observe the inversion of the nature of majority charge carriers (L52). It means that we go from a device in which majority charge carriers are n-type to a device in which majority charge carriers are p-type. We fixed that point all along the manuscript.

  1. The introduction does not sufficiently say that photoactivation depends on the intensity of light. In addition, it is desirable to indicate the intensity of light in the manuscript in the same units of measurement (L211, 219, 234, 240, 243, 320, 347, 368, 486, Table 1).

R: We thank the reviewer for this important remark. This is true, photoactivation depends on the intensity of light. In main text we precised wavelength and light power as much as possible. In the given examples, we added the light power, as soon as it was given in the cited references, and we homogeneized the unit, in mW cm-2.

Additionally, because it is an important parameter, we built a graphical abstract showing that an increase in light power can make the sensor more sensitive to a given target gas, as illustrated by a cat, a dog and an elephant reactions to detection process.

An highlight for this review could be: “An increase in light power can make the sensor more sensitive to a given target gas, as illustrated by a cat, a dog and an elephant reactions to detection process.”

Remarks:

R: We sincerely thank the reviewer for his careful reading and his detailed review.

L8-13. These considerations should be transferred to the Introduction. They are not appropriate in the Annotation.

R: We disagree, so we leave as it was initially.

L39. Gas sensors with a light source do not always have low power consumption, since the light source requires additional energy.

R: This is true, in particular for UV light. So, we added “at least with visible light”.

L52, 600-602,605, 609. The main carriers have a constant and unchanging charge. An inversion of the type of conductivity of a gas-sensitive material, but not of charge carriers, may occur.

R: Thanks to the reviewer. As abovementioned, we replaced by “change in the nature of majority charge carriers”.

L64. Here, photoexcitation of charge carriers, but not of a semiconductor, occurs.

R: We corrected it.

L77-82. The authors should mention here the light intensity for the considered wavelengths.

R: We did it.

L88-89, 93. Phrases are incorrect.

R: We changed the sentence to take into account that the first step is the formation of molecular oxygen, which is adsorbed, but that desorption of O2 occurs in a second step.

L95. The equation is incorrect.

R: We replaced “(g)” by “(ads)” in the equation, taking into account that the first step is the formation of molecular oxygen, which is adsorbed, and that desorption of O2 occurs in a second step.

L126-127. The title of the section inaccurately reflects its content.

R: We changed the title of sections 4 and 5 that become sections 3.1: Chemoresistors and 3.2: Heterojunction Gas Sensors

Fig.2, a. It is necessary to decipher the designations of the abscissa axis.

R: x-Axis is time. However, we deciphered the designation of y-axis in the revised manuscript.

L172. Misspell.

R: We corrected it.

L176-177. Maybe it is necessary to write "with an increase in the concentration of oxygen vacancies"?

R: We corrected it.

L182-184. It is necessary to give the reaction equation.

R: In the cited reference, the mechanism is described in four different steps. We do not feel that it is necessary for the reader, the main idea being that, under light, oxygen desorption induces an increase in oxygen vacancies in these particular materials.

L211, 219, 234, 240, 243, 320, 347, 368, 486, Table.1. It is desirable to indicate the light intensity in the same units of measurement.

R: We did it, choosing mW cm-2.

L260. the abbreviation has not been deciphered.

R: We did it.

L281. "Kelvin" is capitalized.

R: We did it.

L279-284. It is necessary to explain what Kelvin probe force microscopy (KPFM) has to do with the gas sensitivity of the material.

R: Actually, in the present case, Kelvin probe technique gives access to surface potential not to a force; it is not Kelvin probe force microscopy.

L285. This is a repetition of the phrase mentioned above.

R: We modified the text.

L315. What kind of resistor do the authors mean?

R: We precised “ZnO chemoresistors”.

L2, 19,49,140,186, 315 and oth. Instead of resistors, you need to write chemoresistors.

R: We changed “resistors” by “chemoresistors” in most of cases. However, we kept “resistors” in Title and keywords, because in both cases it is to differentiate from heterojunction devices.

L317-321. XPS Peaks of oxygen vacancies are not shown in Fig.5.

R: We moved “Fig. 5” at the good place.

L327. The characterization of CH4 as an explosive gas is superfluous here.

R: We change the few sentences about CH4 detection. With and without oxygen vacancy enrichment, ZnO nanorods (chemoresistor) displayed no gas response to CH4 under both dark and visible light conditions [77]. This observation suggested that oxygen vacancies alone don't contribute to light-activated CH4 sensing at low temperatures. Additionally, it is important to mention that CH4 is a chemically inert gas at room temperature, making it challenging to detect [78].

L353. The expression in parentheses must be given in the form of an equation.

R: Actually, the equation was not necessary and we removed “(CH4  CO2)”

L404, fig.7. What happens to oxygen in the figure? This is not explained anywhere.

R: Recombination of holes and adsorbed anionic oxygen species gives adsorbed O2 molecules, which can then desorb as the form of O2(g), as depicted in equation (1).

L428. The articles [81] and [38] have different authors.

R: We thank the reviewer. Actually, it was not 38 but 36, which is from the same authors as ref. 81.

L449,582, 599.What is "light illumination"?

R: We checked this expression all along the text, and replaced it by “visible light illumination”.

L 462-464. The phrase is incomprehensible. What kind of resistor do the authors mean?

R: We mentioned “with MoS2 and rGO chemoresistors under the same experimental conditions.”

L572-579. Incorrect references to figures are mistakenly indicated in the text.

R: We corrected the citation of Fig. 11 a, b and c.

L573. There is a typo in the name of the heterojunction.

R: Actually, there was no typo, but the initial text was confusing, because we are talking about 4 different heterojunctions, of which only 3 are associated to Fig. 11. We also changed the text to make it clearer:

CuF16Pc/LuPc2 device (Fig. 11c) exhibits positive response to NH3, whereas CuCl8Pc/LuPc2 (Fig. 11a) device exhibits a negative response, as for CuPc/LuPc2 [92]. All these three devices exhibit excellent observable response towards NH3 gas at room temperature in the dark and the response remains almost the same under visible light illumination, while a current increase was observed due to light absorption. Unlike previously discussed sensors, we didn’t observe any enhancement in gas sensing properties upon illumination for the octa-fluorinated copper phthalocyanine-based heterojunction device (CuF8Pc/LuPc2) (Fig. 11b), but the most interesting is the inversion in the nature of the response. So, apart from charge density modification, it was shown for the first time that light can also induce a new type of phenomenon in gas sensors called detrapping effect (Fig. 10c) [18], which can inverse the polarity of majority charge carriers .

L600. What kind of molecule is "dioxygen molecules"?

R: To be coherent all along the text, we changed by “oxygen molecules”

L600-602, 609. The main carriers have a constant and unchanging charge. An inversion of the type of conductivity of the gas-sensitive material may occur, but not an inversion of the charge carriers. The authors mistakenly talk about the inversion of charge carriers. This must be fixed.

R: As answered previously (point 2), we replaced “the inversion of charge carriers” by “the inversion of the nature of majority charge carriers”. With ambipolar materials or devices, we can observe the inversion of the nature of majority charge carriers (L52). It means that we go from a device where majority charge carriers are n-type to a device in which majority charge carriers are p-type. We fixed that point all along the manuscript.

L605, 615. What type of sensors do the authors write about in the manuscript? About resistors, chemosensors, conductometric sensors?

R: This review is devoted to conductometric sensors. We separated our review in chemoresistors and heterostructure gas sensors. To make it clear, in the conclusion we used “on inorganic and organic semiconductor - based chemoresistors and heterojunction gas sensors”

Sincerely

Reviewer 3 Report

Comments and Suggestions for Authors

The review is well written and contains interesting information about the effects of visible light on gas sensors. The review contains explanations of the influence of light and mechanisms, which is very important for articles of this kind.

As a few minor comments, the following can be noted:

1. It is necessary to indicate for what period the articles were reviewed

2. In my opinion, it is necessary to indicate in the text an advantage over existing reviews on similar topics, for example https://doi.org/10.1016/j.ceramint.2020.11.187, https://www.mdpi.com/2072-666X/8/11/333, https://link.springer.com/article/10.1007/s40820-020-00503-4, etc.

3. The conclusion lacks recommendations. What problems exist in this area, advantages and disadvantages of irradiating gas sensors with light?

Author Response

Dear reviewer,

Please find hereafter our response to your comments.

The review is well written and contains interesting information about the effects of visible light on gas sensors. The review contains explanations of the influence of light and mechanisms, which is very important for articles of this kind.

R: We thank the reviewer for the very positive comments.

As a few minor comments, the following can be noted:

  1. It is necessary to indicate for what period the articles were reviewed

R: This review covers mainly articles published since 2000. Actually, the first use of light effect as a tool to improve chemosensors performances started about at that time. Of course, when we are talking about the general effect of light on semiconductors we refer to works published highly before. So, we changed the first sentence of the abstract: Since two decades, many research works were focused on enhancing the properties of gas sensors by utilising external triggers like temperature and light illumination.

  1. In my opinion, it is necessary to indicate in the text an advantage over existing reviews on similar topics, for example https://doi.org/10.1016/j.ceramint.2020.11.187, https://www.mdpi.com/2072-666X/8/11/333, https://link.springer.com/article/10.1007/s40820-020-00503-4, etc.

R: We cited these 3 references in the initial manuscript, and other reviews. However, these reviews focused on inorganic materials, mainly metal oxides and often using UV light. In the present review, we focused on visible light effect, and we extended it to organic materials, as explained in the abstract: “This review summarizes the effect of visible light illumination on both chemoresistors and heterostructure gas sensors based on inorganic and organic materials, and provides a clear understanding about the involved phenomena.”

  1. The conclusion lacks recommendations. What problems exist in this area, advantages and disadvantages of irradiating gas sensors with light?

R: As we answered to reviewer 1 (point 4), this is an important point. We added a few perspectives at the end of the conclusion: Clearly illumination is a wonderful tool that should be more used in conductometric gas sensors. However, wavelength and power light have to be optimized for each sensing material and transducer. In the future, integration of a light source in sensing systems should be as usual as heating systems currently implemented in commercially available gas sensors.

In the introduction, and in the abstract, we mentioned that high energy light can damage sensing materials, which does not occur when visible light is used.

Sincerely

Round 2

Reviewer 1 Report

Comments and Suggestions for Authors

The response and revised manuscript are satisfactory, and it is recommended to accept.

Author Response

Dear reviewer,

We thank you for your acceptance.

Sincerely

Reviewer 2 Report

Comments and Suggestions for Authors

The authors of the manuscript responded in detail to the comments and made the necessary adjustments.

However, in my opinion, the presented graphic abstract does not fully reflect the influence of the radiation intensity. Radiation with a higher intensity can improve the sensitivity of the sensor to the target gas. At the same time, radiation with a higher intensity can destroy the molecules of the target gas (an elephant, cat or dog will become very hot from radiation with a higher intensity).

I believe that after this correction, the manuscript can be accepted for publication.

Author Response

Dear reviewer,

We thank your for your positive comments. About the graphical abstract, everybody knows that it does not contain all the science included in a paper. In contrary, we use graphical abstract to focus on one or a few points. Here, we chose to point the possible response increase under increasing visible light intensity. Even though in some cases an increase in light intensity can cause damages. So, we feel that we have more advantage to save this simple graphical abstract rather than to add information with the risk to dilute our message and to lose the appealing of this picture.

Sincerely
